

# Comparison of circulating tumor cell (CTC) detection rates with epithelial cell adhesion molecule (EpCAM) and cell surface vimentin (CSV) antibodies in different solid tumors: a retrospective study

Yang Gao[1,2], Wan-Hung Fan[3], Zhengbo Song[4], Haizhou Lou[5] and Xixong Kang[1,2]

[1] Key Laboratory for Biomechanics and Mechanobiology of Ministry of Education, School of Biological Science and Medical Engineering, Beihang University, Beijing, China
[2] Beijing Advanced Innovation Center for Biomedical Engineering, Beijing Polytechnic University, Beijing, China
[3] Hangzhou Watson Biotech, Hangzhou, China
[4] Department of Medical Oncology, Zhejiang Cancer Hospital, Hangzhou, China
[5] Department of Oncology, Sir Run Run Shaw Hospital, Zhejiang University School of Medicine, Hangzhou, China

Corresponding author
Xixong Kang, kangxxtt@sina.com

## ABSTRACT

**Purpose:** Status of epithelial-mesenchymal transition (EMT) varies from tumors to tumors. Epithelial cell adhesion molecule (EpCAM) and cell surface vimentin (CSV) are the most common used targets for isolating epithelial and mesenchymal CTCs, respectively. This study aimed to identify a suitable CTC capturing antibody for CTC enrichment in each solid tumor by comparing CTC detection rates with EpCAM and CSV antibodies in different solid tumors.

**Methods:** Treatment-naive patients with confirmed cancer diagnosis and healthy people who have performed CTC detection between April 2017 and May 2018 were included in this study. CTC detection was performed with CytoSorter® CTC system using either EpCAM or CSV antibody. In total, 853 CTC results from 690 cancer patients and 72 healthy people were collected for analysis. The performance of CTC capturing antibody was determined by the CTC detection rate.

**Results:** EpCAM has the highest CTC detection rate of 84.09% in CRC, followed by BCa (78.32%). CTC detection rates with EpCAM antibody are less than 40% in HCC (25%), PDAC (32.5%) and OC (33.33%). CSV has the highest CTC detection rate of 90% in sarcoma, followed by BC (85.71%), UC (84.62%), OC (83.33%) and BCa (81.82%). CTC detection rates with CSV antibody are over 60% in all 14 solid tumors. Except for CRC, CSV has better performances than EpCAM in most solid tumors regarding the CTC detection rates.

**Conclusion:** EpCAM can be used as a target to isolate CTCs in CRC, LC, GC, BCa, EC, HNSCC, CC and PCa, especially in CRC, while CSV can be used in most solid tumors for isolating CTCs.

# INTRODUCTION

Circulating tumor cells (CTCs) are tumor cells that have shed from tumor tissues and escaped into circulation (*Lozar et al., 2019*). CTCs are circulating within the bloodstream and can recolonize a distant site under a suitable condition, a phenomenon called metastasis, which is the main cause of death in most cancer patients (*Lozar et al., 2019*). CTCs represent the undergoing process of metastasis and therefore American Joint Committee on Cancer (AJCC) has introduced a new cancer stage, cM0 (i+), in the 7th edition of Staging Manual for breast cancer (BCa) (*Edge & Compton, 2010*). cM0 (i+) is defined as a tumor stage when no clinical or radiographic evidence of distant metastases is found, but tumors cells are still detected in the bone marrow, blood or distant non-regional lymph nodes (*Edge & Compton, 2010*). The clinical values of CTCs have been written in the Chinese expert consensus in lung cancer (LC), colorectal cancer (CRC), BCa, esophageal cancer (EC), hepatocellular carcinoma (HCC) and pancreatic cancer (*Chinese Society of Clinical Oncology, 2018*; *Union of Chinese Oncology Management, 2019*; *Chinese Research Hospital Association, 2019a*, *2019b*; *Shandong Research Hospital Association, 2019*; *Chinese Medical Association, 2020*), and demonstrated in the prediction of patients' prognosis, monitoring of tumor recurrence, cancer diagnosis and screening, evaluation of the treatment responses, and guidance of treatment (*Lozar et al., 2019*).

It is estimated that $3.2 \times 10^6$ tumor cells detach from one gram of tumor tissue per day (*Butler & Gullino, 1975*), but most of them quickly proceed to apoptosis due to the loss of adhesion to the extracellular matrix, hemodynamic shear forces, or attacks from the immune systems (*Lozar et al., 2019*; *Butler & Gullino, 1975*), which contributes to the first characteristics of CTCs, the rarity. The second characteristic of CTCs is the heterogeneity. Even CTCs from the same patients may differ in terms of cell size, morphology, and gene expression. An ideal CTC detection platform allows for enrichment of all heterogeneous CTCs, while discarding the majority of the surrounding blood cells. Thus, the first step of CTC detection is to isolate CTCs from the background of blood cells, followed by an identification step that can further distinguish (and possibly characterize) CTCs from the remaining cells. Many techniques have been developed to enrich CTCs, depending on either the unique biophysical or biochemical properties of CTCs (*Tellez-Gabriel, Heymann & Heymann, 2019*). Biophysical methods, also known as label-free methods, separate CTCs by size, density, electric charge, and deformability, while biochemical methods rely on the recognition of different surface biomarkers between CTCs and leukocytes. An ideal biomarker for CTCs isolation should be some surface molecules expressed universally and only on CTCs. Unfortunately, such molecule is not yet found. The most commonly used target for positive immuno-selection of CTCs is the epithelial cell adhesion molecule (EpCAM) (*Lozar et al., 2019*), a cell surface glycoprotein typically over-expressed in epithelial cancer cells. EpCAM is not expressed on

tumors of mesodermal and ectodermal origin, such as neurogenic tumors, sarcomas, melanomas, or lymphomas (*Trzpis et al., 2007*). As the first and only US Food and Drug Administration (FDA) approved CTC detection technique, CellSearch® uses EpCAM antibody functionalized magnetic beads and cytokeration (CK) antibody to capture and identify CTCs, respectively (https://www.cellsearchctc.com/). A major drawback of EpCAM based methods is their inability to detect EpCAM negative CTCs. Epithelial-mesenchymal transition (EMT) is a cellular process in which adherent epithelial cells acquire a migratory mesenchymal phenotype, and usually observed during morphogenesis at the time of embryonic development (*Yang & Weinberg, 2008*). Studies have shown that EMT is quite common in tumor and plays an important role during metastasis, allowing tumor cells to become more aggressive, chemo-resistant and invasive (*Yang & Weinberg, 2008*; *Satelli et al., 2015*; *Kalluri, 2009*). CTCs that have undergone EMT survive better in the circulation. EMT process is usually accompanied by the over-expression of mesenchymal markers, such as vimentin, N-cadherin and Twist, while reducing expression of epithelial makers, such as EpCAM, cytokeratin (CK) and E-cadherin (*Chaw et al., 2012*). Thus, EMT induces the generation of EpCAM-negative CTCs. In fact, CTCs at different EMT states could be found in the circulation (*Pastushenko & Blanpain, 2019*; *Yu et al., 2013*). EpCAM antibody might not be able to capture the EpCAM-negative CTCs, which should be the majority of surviving CTCs in the circulation.

Vimentin, a 57kDa intermediate filament protein, is a major cytoskeleton component of mesenchymal cells (*Satelli & Li, 2011*). As EMT of cancer cells induces the over-expression of vimentin and translocation of vimentin to the cell surface (*Satelli & Li, 2011*), Satelli et al tried to raise an antibody against cell surface vimentin (CSV) to capture the EMT type of CTCs (*Satelli et al., 2014*). CTCs captured by CSV antibody expressed other EMT markers as well, such as Snail, Slug and Twist, suggesting that CSV could be used as a marker to isolate EMT type of CTCs (*Satelli et al., 2014*). Uses of CSV antibody to capture mesenchymal CTCs have been validated in breast, prostate, colorectal and pancreatic cancers (*Satelli et al., 2015*; *Satelli et al., 2017*; *Satelli et al., 2016*; *Wei et al., 2019*). Mesenchymal CTCs are believed to have more clinical significance. The presence of mesenchymal CTCs is usually associated with disease progression and worse prognosis (*Jolly et al., 2019*). In a comparative study between early and metastatic patients of BCa, mesenchymal CTCs were identified in 73% and 100% of patients (*Kallergi et al., 2011*). Furthermore, expression of mesenchymal markers correlates with lymph node involvement. Taken together, it, suggests that the EMT phenotype is directly related to the metastatic potential of CTCs (*Markiewicz et al., 2013*).

Circulating tumor cells are a collection of heterogeneous cells, comprising either epithelial or mesenchymal or intermediate phenotype (*Pastushenko & Blanpain, 2019*; *Yu et al., 2013*). *Tan et al. (2014)* showed that every cancer has an unique cancer-specific EMT signature and EMT spectrums varied between different cell lines and tumors. EpCAM and CSV can be used as targets to isolate the epithelial and mesenchymal subtypes of CTCs, respectively. Studies in breast and pancreatic cancers have shown the use of CSV is more efficient to capture CTCs than EpCAM (*Satelli et al., 2015*; *Wei et al., 2019*).

This study aimed at defining a suitable CTC enrichment antibody for each solid tumor by comparing the CTC detection rates with EpCAM or CSV antibodies in different solid tumors.

## METHODS AND MATERIALS

### Patients

It is a retrospective study. Cancer patients or healthy individuals who have performed CTC detection between April 2017 and May 2018 in Zhejiang University Medical College Affiliated Sir Run Run Shaw Hospital, Zhejiang University Medical College Second Affiliated Hospital and Zhejiang Cancer Hospital were included in the study. Inclusion criteria for cancer patients were as follows: (1) patients were diagnosed with confirmed cancer; (2) patients were treatment-naive before CTC detection; (3) patients had negative history of other malignancy within 5 years prior CTC detection. Patients of the following descriptions were excluded and rejected from the study: (1) patients were pregnant or breast-feeding at time of CTC detection or surgery; (2) patients with more than one type of cancer; (3) patients had other conditions which investigators thought was not suitable for the study. Healthy people were defined as people without any clinical symptoms of any disease at time time of CTC detection. In total, 72 healthy people and 690 cancer patients were recruited, comprising 140 LC, 85 CRC, 92 BCa, 70 gastric cancer (GC), 66 pancreatic ductal adenocarcinoma (PDAC), 50 cervical cancer (CC), 54 with head and neck squamous cell carcinoma (HNSCC), 31 EC, 21 brain cancer (BC), 19 HCC, 25 prostate cancer (PCa), 14 ovarian cancer (OC), 13 bladder cancer (UC), and 10 sarcoma. Detailed participant information was summarized in the Table S1. This study followed the principles established in the Declaration of Helsinki and was approved by the ethics committee of each participated hospitals with IRB number, Qi Xie Lin Chuang Shi Yan 20180427-1, (2018)Lun Shen Shi Ji Di(013)Hao, and 2015-01-45 Hao, respectively.

### CTC detection

A total of 853 CTC results from 690 cancer patients and 72 healthy people were collected. A total of 341 and 512 people had CTC detection for epithelial and mesenchymal CTCs, respectively, while 91 cancer patients had CTC detection for both epithelial and mesenchymal CTCs at same time. CTCs were detected by CytoSorter® (Hangzhou Watson Biotech, Hangzhou, China), using CytoSorter® circulating epithelial cells detection kit (EpCAM) and/or mesenchymal cells detection kit (CSV). Biotin-labeled CTC capturing antibodies were anti-EpCAM monoclonal antibody (Biotin) (DH0023, Abnova, Taiwan) and cell-surface vimentin (CSV) monoclonal antibody, clone 84-1 (Biotin) (H00007431-MB08, Abnova, Taiwan). CytoSorter® technology employs the positive selection for CTCs by utilizing a streptavidin nanoarray on a Chip, CytoChipNano, which can be then coated with the biotin-labeled capturing antibody, and immunofluorescence staining, in order to isolate and identify CTCs. CytoSorter® can be used with any biotin-labeled antibody to capture desired targeted CTCs. EpCAM and CSV antibodies were coated on the chips separately to enrich the epithelial and mesenchymal types of CTCs, respectively. Epithelial CTCs are identified as PanCK-positive, CD45-negative and DAPI-positive

**Table 1 CTC detection rates in different solid cancers with EpCAM or CSV antibody.**

| Cancer Type | EpCAM | | | | CSV | | | |
|---|---|---|---|---|---|---|---|---|
| | *n* | CTC Detected | Detection Rate (%) | CTCs Range (/4 mL) | *n* | CTC Detected | Detection Rate (%) | CTCs Range (/4 mL) |
| LC | 62 | 38 | 61.29 | 0–24 | 98 | 76 | 77.55 | 0–22 |
| CRC | 44 | 37 | 84.09 | 0–20 | 49 | 33 | 67.35 | 0–9 |
| PDAC | 40 | 13 | 32.50 | 0–3 | 46 | 31 | 67.39 | 0–18 |
| GC | 29 | 15 | 51.72 | 0–13 | 51 | 32 | 62.75 | 0–8 |
| BCa | 47 | 38 | 78.32 | 0–15 | 55 | 45 | 81.82 | 0–8 |
| HNSCC | 24 | 16 | 66.67 | 0–36 | 30 | 21 | 70.00 | 0–12 |
| HCC | 4 | 1 | 25.00 | 0–5 | 19 | 14 | 73.68 | 0–14 |
| CC | 29 | 19 | 65.52 | 0–8 | 29 | 22 | 75.86 | 0–7 |
| EC | 14 | 8 | 57.14 | 0–8 | 21 | 15 | 71.43 | 0–7 |
| OC | 6 | 2 | 33.33 | 0–3 | 12 | 10 | 83.33 | 0–7 |
| Sarcoma | 3 | 0 | 0.00 | 0 | 10 | 9 | 90.00 | 0–9 |
| BC | 0 | N/A | N/A | N/A | 21 | 18 | 85.71 | 0–6 |
| UC | 0 | N/A | N/A | N/A | 13 | 11 | 84.62 | 0–3 |
| PCa | 16 | 10 | 67.74 | 0-8 | 9 | 6 | 66.67 | 0–11 |
| Healthy Control | 23 | 2 | 8.51 | 0-1 | 49 | 5 | 10.20 | 0–1 |

Note:
EpCAM, epithelial cell adhesion molecule; CSV, cell surface vimentin; *n*, sample number; CTC, circulating tumor cell; LC, lung cancer; CRC, colorectal cancer; BCa, breast cancer; GC, gastric cancer; PDAC, pancreatic ductal adenocarcinoma; CC, cervical cancer; HNSCC, head and neck squamous cell carcinoma; EC, esophageal cancer; BC, brain cancer; N/A, not available; HCC, hepatocellular carcinoma; PCa, prostate cancer; OC, ovarian cancer; UC, bladder cancer.

cells, while mesenchymal CTCs are identified as CSV-positive, CD45-negative and DAPI-positive cells. CTC detection procedure was following the manufacturer protocol and was described in the previous study (*Wei et al., 2019*; *Zheng et al., 2019*). An epithelial CTC was identified as a PanCK-FITC+, CD45-PE−, and DAPI+ cell, while a mesenchymal CTC was identified as a CSV-FITC+, CD45-PE−, and DAPI+ cell.

## Statistical analysis

As recruited patients were at different cancer stages with different solid tumors, only CTC detection rates, were analyzed among different cancers. Only CTC enumerations from the same patients' blood drawings with different capturing antibodies would be compared. Statistical analysis was performed using Prism 5.0 (Graphpad, La Jolla, CA, USA). Analysis of two groups of data for the differences between the enumerations was determined using a student's *t* test. A two-sided *p* value less than 0.05 was considered statistically significant.

## RESULTS

### Identification of epithelial CTCs in different solid tumors

As shown in Table 1, 341 CTC results for epithelial CTC, 318 cancer patients with 12 different solid tumors and 23 healthy people, were included for analysis. An epithelial CTC is shown in Fig. 1A as indicated by the yellow arrow. CTCs were detected in 197

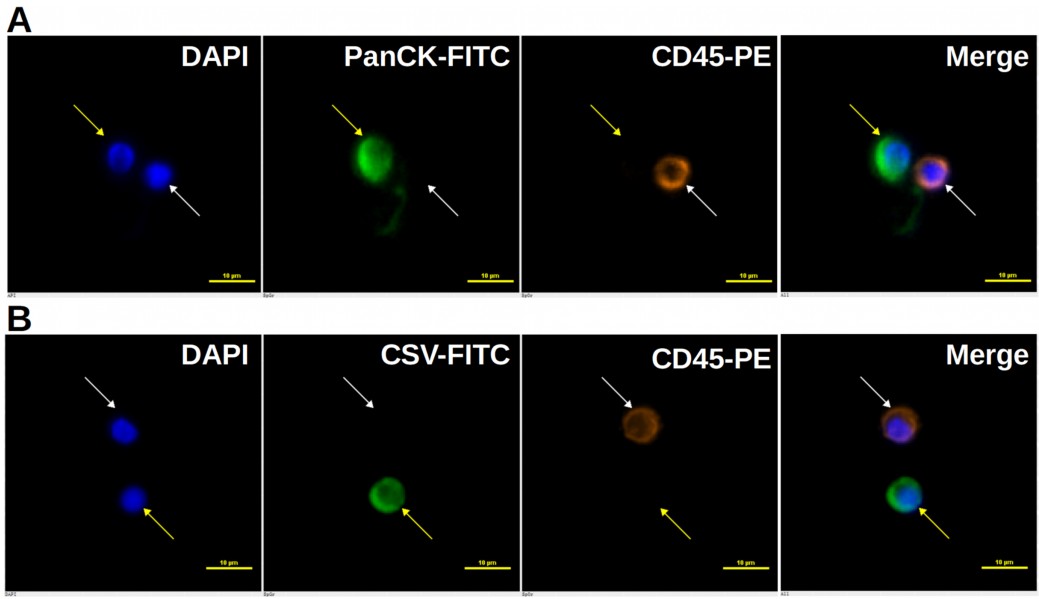

**Figure 1 Immunofluorescent staining of CTCs.** Immunofluorescent staining of CTCs. (A) An epithelial CTC is identified as a DAPI (blue) positive, PanCK (FITC, green) positive and CD45 (PE, orange) negative cell as indicated by the yellow arrow. (B) A mesenchymal CTC is identified as a DAPI (blue) positive, CSV (FITC, green) positive and CD45 (PE, orange) negative cell as indicated by the yellow arrow. White blood cells, marked by white arrows, are identified as DAPI (blue) positive, FITC (green) negative and CD45 (PE, orange) positive cells. Scale bar represents 10 μm, immunofluorescent staining, × 20.              

out of 318 cancer patients, and 2 out of 23 healthy people as shown in Table 1. Healthy people usually have CTCs no more than 2, while cancer patients have usually CTCs more than 1. Epithelial CTCs detected by CytoSorter® might be used as a diagnostic aid for cancer screening with a sensitivity and specificity of 0.619 and 0.913, respectively. Average CTC detection rate with EpCAM is 61.95%. No epithelial CTC was found in sarcoma patients, which is in accordance with the expectation since sarcoma is a mesenchymal type of tumors. As shown in Fig. 2A, EpCAM has the highest CTC detection rate of 84.09% in CRC, followed by BCa (78.32%). EpCAM has CTC detection rates less than 50% in HCC (25%), PDAC (32.5%) and OC (33.33%). The CTC detection rate is not associated with patients' gender or age (both $P > 0.05$).

## Identification of mesenchymal CTCs in different solid tumors

512 CTC results for mesenchymal CTCs, including 463 cancer patients with 14 different solid tumors and 49 healthy people, we collected for analysis. A mesenchymal CTC is shown in Fig. 1B as indicated by the yellow arrow. Mesenchymal CTCs were detected in 325 out of 463 cancer patients, and in 5 out of 49 healthy people as shown in Table 1. Like the EpCAM-based kit, healthy people have a maximum of 1 CTC detected, while cancer patients have usually more than 1 CTCs. Thus, it indicates as well that mesenchymal CTCs captured by CSV antibody might be used as a screening aid for cancers with a sensitivity and specificity of 0.702 and 0.898, respectively. CSV has the highest CTC detection rate of 90% in the mesenchymal cancer, sarcoma. Beside sarcoma,
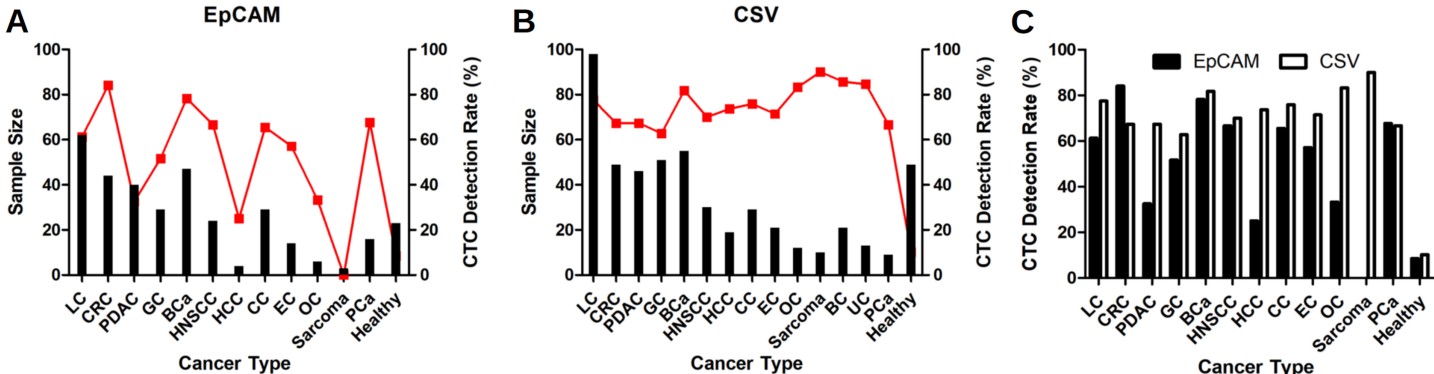

**Figure 2 CTC detection rates in different solid tumors.** CTC detection rates in different solid tumors with EpCAM antibody as shown in (A), or with CSV antibody as shown in (B). Red dots represent the CTC detection rates and black bars represent the sample size. (C) Comparison of CTC detection rate with EpCAM and CSV antibodies in different solid tumors. LC, lung cancer; CRC, colorectal cancer; BCa, breast cancer; GC, gastric cancer; PDAC, pancreatic ductal adenocarcinoma; CC, cervical cancer; HNSCC, head and neck squamous cell carcinoma; EC, esophageal cancer; BC, brain cancer; HCC, hepatocellular carcinoma; PCa, prostate cancer; OC, ovarian cancer; UC, bladder cancer.

CSV has the highest CTC detection rate of 85.71% in BC, followed by UC (84.62%), OC (83.33%) and BCa (81.82%). As shown in Fig. 2B, CTC detection rates with CSV in all 14 solid tumors are all over 60%, with an average detection rate of 70.19%. The CTC detection rate is not associated with patients' gender or age (both $P > 0.05$).

## Comparison of CTC detection rates with EpCAM or CSV antibody in 12 different solid tumors

As shown in Table 1 and Fig. 2C, CSV has better performances than EpCAM in most cancers in regarding CTC detection rates, except for CRC. In CRC, EpCAM has a higher CTC detection rate of 84.09% than CSV (67.35%). CSV shows much better performances of 67.39%, 73.68% and 83.33%, than EpCAM (32.5%, 25% and 33.33%, respectively) in PDAC, HCC and OC. In LC and EC, CSV performs slightly better than EpCAM (77.55% vs 61.29% and 71.43% vs 57.14%, respectively). In BCa, GC, HNSCC, CC and PCa, CSV and EpCAM have similar performances.

## Comparison of CTC enumerations with EpCAM and CSV antibodies from the same blood drawings in the same patients

A total of 91 recruited cancer patients had performed CTC detection by EpCAM and CSV antibodies at same time, including 20 LC, 8 CRC, 20 PDAC, 10 GC, 10 BCa, 4 HCC, 8 CC, 4 EC, 4 OC and 3 sarcoma. Results are shown in Table 2 and Fig. S1. CTC detection rates with CSV are higher than those with EpCAM. Although there is no statistical significance, patients with more epithelial CTCs tend to have more mesenchymal CTCs. CSV is more efficient to capture CTCs in PDAC patients than EpCAM as shown in Table 2 ($P = 0.0011$). As for other cancers, no correlation is found between CTC enumerations by EpCAM and CSV.

**Table 2 Comparison of CTC enumerations with EpCAM and CSV antibodies in the same patients.**

| Patient No. | CTC Count EpCAM | CSV | P Value | Patient No. | CTC Count EpCAM | CSV | P Value | Patient No. | CTC Count EpCAM | CSV | P Value |
|---|---|---|---|---|---|---|---|---|---|---|---|
| LC (n = 20) | | | | PDAC (n = 20) | | | | BCa (n = 10) | | | |
| 1 | 0 | 3 | 0.1283 | 29 | 0 | 2 | 0.0011 | 59 | 6 | 10 | 0.9051 |
| 2 | 0 | 0 | | 30 | 1 | 4 | | 60 | 5 | 6 | |
| 3 | 0 | 5 | | 31 | 0 | 3 | | 61 | 0 | 1 | |
| 4 | 0 | 4 | | 32 | 0 | 3 | | 62 | 9 | 6 | |
| 5 | 0 | 3 | | 33 | 1 | 0 | | 63 | 0 | 3 | |
| 6 | 5 | 5 | | 34 | 0 | 7 | | 64 | 4 | 0 | |
| 7 | 1 | 2 | | 35 | 3 | 11 | | 65 | 3 | 1 | |
| 8 | 4 | 2 | | 36 | 1 | 0 | | 66 | 0 | 1 | |
| 9 | 5 | 3 | | 37 | 3 | 11 | | 67 | 3 | 1 | |
| 10 | 7 | 11 | | 38 | 0 | 4 | | 68 | 0 | 0 | |
| 11 | 24 | 16 | | 39 | 0 | 7 | | HCC (n = 4) | | | |
| 12 | 12 | 13 | | 40 | 0 | 1 | | 69 | 0 | 0 | 0.5862 |
| 13 | 8 | 9 | | 41 | 0 | 8 | | 70 | 0 | 2 | |
| 14 | 18 | 3 | | 42 | 0 | 2 | | 71 | 0 | 1 | |
| 15 | 2 | 4 | | 43 | 2 | 1 | | 72 | 5 | 4 | |
| 16 | 18 | 22 | | 44 | 0 | 17 | | CC (n = 8) | | | |
| 17 | 20 | 22 | | 45 | 0 | 7 | | 73 | 0 | 1 | 0.7995 |
| 18 | 11 | 14 | | 46 | 1 | 4 | | 74 | 9 | 2 | |
| 19 | 5 | 5 | | 47 | 3 | 2 | | 75 | 1 | 2 | |
| 20 | 0 | 4 | | 48 | 2 | 1 | | 76 | 0 | 2 | |
| CRC (n = 8) | | | | GC (n = 10) | | | | 77 | 1 | 4 | |
| 21 | 0 | 3 | 0.2065 | 49 | 0 | 4 | 0.2312 | 78 | 5 | 1 | |
| 22 | 2 | 0 | | 50 | 0 | 4 | | 79 | 2 | 2 | |
| 23 | 4 | 0 | | 51 | 0 | 2 | | 80 | 0 | 6 | |
| 24 | 0 | 2 | | 52 | 0 | 1 | | EC (n = 4) | | | |
| 25 | 20 | 9 | | 53 | 0 | 1 | | 81 | 1 | 2 | 0.1489 |
| 26 | 11 | 5 | | 54 | 0 | 0 | | 82 | 0 | 1 | |
| 27 | 12 | 2 | | 55 | 5 | 4 | | 83 | 0 | 1 | |
| 28 | 0 | 3 | | 56 | 3 | 1 | | 84 | 3 | 3 | |
| | | | | 57 | 4 | 2 | | OC (n = 4) | | | |
| Sarcoma (n = 3) | | | | 58 | 1 | 5 | | 85 | 3 | 3 | 0.5 |
| 89 | 0 | 3 | 0.25 | | | | | 86 | 0 | 2 | |
| 90 | 0 | 5 | | | | | | 87 | 0 | 1 | |
| 91 | 0 | 9 | | | | | | 88 | 1 | 1 | |

Note:
CTC, circulating tumor cell; EpCAM, epithelial cell adhesion molecule; CSV, cell surface vimentin; No, number; n, sample number; LC, lung cancer; CRC, colorectal cancer; BCa, breast cancer; GC, gastric cancer; PDAC, pancreatic ductal adenocarcinoma; CC, cervical cancer; HNSCC, head and neck squamous cell carcinoma; EC, esophageal cancer; BC, brain cancer; N/A, not available; HCC, hepatocellular carcinoma; PCa, prostate cancer; OC, ovarian cancer; UC, bladder cancer.
## DISCUSSIONS

Our results show that both EpCAM and CSV have different CTC detection rates among different solid tumors. Especially for EpCAM, the rates vary a lot from 84.09% in CRC to 25% in HCC. CTC detection rates with EpCAM antibody are quite in accordance with EpCAM expression profiles in different cancers. According to the Cancer Genome Atlas (TCGA) dataset, EpCAM is expressed in the cytoplasm and cell membranes of glandular cells, and most abundant in the gastrointestinal tract (colon, rectum and gallbladder) and thyroid gland as shown in Fig. S2 (https://www.proteinatlas.org/ENSG00000119888-EPCAM/tissue). Furthermore, CRC has the lowest EMT score among all cancers (*Tan et al., 2014*). Taken together, the high expression of EpCAM in colon and rectum and low EMT in CRC might explain why EpCAM has the highest CTC detection rate in CRC (*Tan et al., 2014*; https://www.proteinatlas.org/ENSG00000119888-EPCAM/tissue; *Spizzo et al., 2011*). On the contrary, there is a very low expression of EpCAM in the ovary and liver (https://www.proteinatlas.org/ENSG00000119888-EPCAM/tissue; *Spizzo et al., 2011*). Therefore, EpCAM have lower CTC detection rates of 33.33% and 25% in OC and HCC, respectively. EpCAM is moderately expressed in lung, nasopharynx, breast, cervix, male organs and pancreas (https://www.proteinatlas.org/ENSG00000119888-EPCAM/tissue), and thus CTC detection rates are ranged mostly from 50% to 70% in these relevant solid tumors. Studies have shown that more mesenchymal CTCs would be found in patients at advanced cancer stage (*Kallergi et al., 2011*) and patients with more mesenchymal CTCs have higher metastasis potentials (*Yu et al., 2013*). Although EpCAM is moderately expressed in pancreas, the CTC detection rate is only 32.5%. It may be due to that most PDAC patients enrolled in this study were already at advanced cancer stage at time of diagnosis.

Due to the vimentin's unique characteristics of translocating from intracellular region to the cell surface in cells undergoing EMT (*Mitra et al., 2015*), CSV has been proposed to be the target for the isolation of mesenchymal CTCs (*Satelli et al., 2014*). Our results show that the use of CSV antibody to detect CTCs would generate higher CTC detection rates in most solid tumors compared to the EpCAM, which is consistent with previous findings in BCa and PDAC (*Satelli et al., 2015*; *Wei et al., 2019*). Furthermore, CTC detection rates with CSV are all over 60% in general. Regarding the CTC detection rates, CSV performs much better than EpCAM in PDAC, HCC and OC. As reported by the TCGA dataset, HCC has low, while OC has high EpCAM expression (https://www.proteinatlas.org/ENSG00000119888-EPCAM/tissue). However, both have high EMT scores (*Tan et al., 2014*), indicating most cancer cells from HCC and OC would undergo EMT, which might explain why CSV antibody performs better in these two cancers.

Although EpCAM is highly expressed in gastrointestinal tract (https://www.proteinatlas.org/ENSG00000119888-EPCAM/tissue), EpCAM has a CTC detection rate of only 51.72% in GC, where CSV also has a low CTC detection rate of 62.75%. GC has a low EMT score and hematogenous metastasis is a rare phenomenon in early stage of GC, which might explain the overall low CTC detection rates by both antibodies in GC.

We compared the CTC counts with both EpCAM and CSV methods from the same blood drawing in 91 cancer patients and only results from the PDAC patients show statistical significance. It may be explained by the limited sample sizes or that PDAC patients recruited were at advanced cancer stage while patients of other cancers were at mixed cancer stages.

Both EpCAM and CSV can generate false-positive results in healthy controls, but CTC counts in healthy controls are usually less than 2. The reason for the false-positive results with EpCAM method might be due to the potential skin epithelial cell contamination from venipuncture. CSV cannot only be seen in EMT type of cancer cells, but also in apoptotic neutrophils, some virus-infected cells and in myofibroblasts and activated stellate cells (*Moisan & Girard, 2006*; *Du et al., 2014*; *Song & Ise, 2020*). CSV expression is shown to have correlations with metastatic phenotype in aggressive cancer and sarcoma (*Satelli & Li, 2011*; *Satelli et al., 2014*; *Satelli et al., 2015*). Expression of CSV on activated macrophages, platelets and apoptotic T lymphocytes evidenced its participation in human neutrophil spontaneous apoptosis and further corroborated its association with inflammatory diseases and signaling (*Moisan & Girard, 2006*). As a putative anti-viral drug target, CSV takes part in viral multiplication by facilitating the internalization of virions (*Du et al., 2014*). Therefore, false-positive results with CSV method might be observed in people with inflammatory diseases, virus infection or fibrosis lesion (*Moisan & Girard, 2006*; *Du et al., 2014*; *Song & Ise, 2020*). Vimentin is normally expressed in mesenchymal cells, such as white blood cells (WBC), as a component of cytoskeleton (*Satelli & Li, 2011*). A damaged WBC might also lead to false-positive results.

Although EMT can increase tumor cells' invasiveness, promote tumor cells' intravasation and ensure tumor cells' survival in the circulation (*Jie, Zhang & Xu, 2017*), the role of mesenchymal CTCs in metastasis remains under debate (*Garber, 2008*; *Chui, 2013*). Some believe that EMT is of great importance in the formation of metastases (*Satelli et al., 2015*; *Pastushenko & Blanpain, 2019*), while others think EMT might have been overestimated and have suggested that EMT is not required for metastasis (*Garber, 2008*; *Chui, 2013*). A recent in vivo animal study demonstrated that epithelial CTCs with restricted mesenchymal transition are a major source of metastases in BCa (*Liu et al., 2019*). Still more studies support that mesenchymal CTCs are more important to predict patients' survival outcomes (*Pastushenko & Blanpain, 2019*; *Hou, Guo & Lyu, 2019*; *Yang et al., 2019*). But evidences suggest that poor prognosis of patients with more mesenchymal CTCs might be due to the increased chemoresistance rather than metastasis potential (*Fischer et al., 2015*; *Zheng et al., 2015*). The EMT spectrums differ from tumors to tumors (*Tan et al., 2014*). A low EMT score does not indicate a restricted metastasis potential. Although CRC has the lowest EMT score, it can still easily develop metastasis. Each cancer may use different strategies to proceed metastasis.

This study aimed to identify a suitable CTC capturing antibody in each solid tumor. The major drawbacks of this study are the fact that it is a retrospective study, the limited sample size and capturing antibodies tested. Comparison of CTC detection rates with different CTC capturing antibodies should be performed in the same subjects to avoid the personal interference. Since it is a retrospective study, we were only able to collect

91 subjects who had CTC detection with both EpCAM and CSV antibodies. Only LC, BCa, and GC have sample sizes more than 50, and no patients with benign tumors were included for the analyses. Due to the small sample size, no correlation of CTCs with patients' pathological characteristics is found in most solid tumor. As there is no universal biomarker for cancer, we compared the CTC detection efficiency in each solid tumor with EpCAM or CSV antibody. Although EpCAM and CSV are considered as common targets for epithelial and mesenchymal CTCs, respectively, antibodies against tissue specific antigen, such as human epidermal growth factor receptor 2 (HER2) for BCa and prostate specific membrane antigen (PSMA) for PCa, should be also included in this study. CTCs are heterogeneous, therefore, no single CTC capturing antibody is able to isolate all CTCs. Combination of multiple capturing antibodies should increase the CTC detection efficiency. It is reported that combined antibodies against folic acid (FA), epidermal growth factor receptor (EGFR) and vimentin can increase the CTC detection rate in LC (*Li et al., 2020*). Thus, combination of multiple CTC capturing antibodies should be also included in this study.

Our results show that EpCAM and CSV antibodies have different CTC detection rates across different tumors. Both EpCAM and CSV are not perfect markers for the isolation of CTCs. EpCAM is not expressed in every epithelial tumor (https://www.proteinatlas.org/ENSG00000119888-EPCAM/tissue) and may be lost in cells undergoing EMT (*Chaw et al., 2012*). CSV is expressed in EMT cells, apoptotic neutrophils, virus-infected cells and myofibroblasts (*Satelli & Li, 2011*; *Moisan & Girard, 2006*; *Du et al., 2014*; *Song & Ise, 2020*). The use of CSV to detect CTCs in the blood may generate some false-positive results. Since there is currently no other better markers, both are acceptable to detect CTCs in certain tumors. Based on our results, EpCAM can be used to isolate CTCs in EpCAM highly expressing tumors and in tumors with low EMT scores, such as CRC, LC, GC, BCa, EC, HNSCC, CC and PCa. CSV can be generally used to enrich CTCs in most solid tumors with acceptable CTC detection rates more than 60%.

## CONCLUSIONS

The results of this study showed that EpCAM can be used as a target to isolate CTCs in CRC, LC, GC, BCa, EC, HNSCC, CC and PCa, but not suitable in PDAC, HCC and OC, while CSV can be used in most solid tumors. However, these results should be prospectively validated with bigger sample sizes.

### Funding
The authors received no funding for this work.

### Competing Interests
Wan-Hung Fan is currently employed by Hangzhou Watson Biotech Inc. The authors have declared that they have no competing interests.

## Author Contributions

- Yang Gao conceived and designed the experiments, performed the experiments, analyzed the data, prepared figures and/or tables, and approved the final draft.
- Wan-Hung Fan conceived and designed the experiments, performed the experiments, analyzed the data, prepared figures and/or tables, and approved the final draft.
- Zhengbo Song conceived and designed the experiments, authored or reviewed drafts of the paper, and approved the final draft.
- Haizhou Lou conceived and designed the experiments, authored or reviewed drafts of the paper, and approved the final draft.
- Xixong Kang conceived and designed the experiments, authored or reviewed drafts of the paper, and approved the final draft.

## Human Ethics

The following information was supplied relating to ethical approvals (i.e., approving body and any reference numbers):

This study followed the principles established in the Declaration of Helsinki and was approved by the ethics committee of Zhejiang University Medical College Affiliated Sir Run Run Shaw Hospital, Zhejiang University Medical College Second Affiliated Hospital and Zhejiang Cancer Hospital with IRB number, Qi Xie Lin Chuang Shi Yan 20180427-1, (2018) Lun Shen Shi Ji Di (013) Hao, and 2015-01-45 Hao, respectively.

## Data Availability

Raw data is available in the Supplemental Files.

## Supplemental Information

Supplemental information for this article can be found online at http://dx.doi.org/10.7717/peerj.10777#supplemental-information.

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
