# Peer review of "Comparison of circulating tumor cell (CTC) detection rates with epithelial cell adhesion molecule (EpCAM) and cell surface vimentin (CSV) antibodies in different solid tumors: a retrospective study"

_PeerJ, doi:10.7717/peerj.10777_

## Round 0.1 · original submission · Major Revisions

The reviewers have several suggestions to improve your manuscript in order to reach the quality necessary and even provide you an annotated manuscript. Reviewer 3 suggests adding a figure after datamining TCGA for EpCAM and CSV expression, which is an additional good idea.

Please let us know if you need more time to revise your manuscript if feasible for you to address the point of tumor origin.

Reviewer 1 ·

Basic reporting

no comment

Experimental design

no comment

Validity of the findings

no comment

Additional comments

Comments:
This is an interesting study investigating the CTC detection rates with EpCAM and CSV capture antibodies in different solid tumors. The findings provide evidence that EpCAM can be used in CRC, LC, GC, BCa, EC, HNSCC, CC and PCa, but not suitable in PDAC, HCC and OC, while CSV can be used in most solid tumors for CTC detection.
The manuscript is well designed and the analyses are sufficiently performed. I have some comments to mention:
Question 1: The method and result part are not clear enough to read and to understand. The authors should revise these parts. It should be easily to read and give the reader the chance to understand.
Question 2: The authors should describe clearly that how many patients were captured CTCs with EpCAM and CSV antibodies separately and both? The numbers in the manuscript are very confusing. Please make it clear.
Question 3: The patients’ clinic-pathological characteristics are described poorly. The authors should give an exact overview of disease histology, grade, stage, the types of surgery, and IHC staining data and disease status.
Question 4: The authors should provide the patients’ status at the time of blood draw, pre- or post-surgery, pre- or post-chemotherapy or immune therapy, first-line or second line therapy, et al.
Question 5: The authors should describe the method of data collection in method and material parts, including clinical data collection and blood sample collection.
Question 6: The authors should provide the inclusion and exclusion criteria of enrolled patients and healthy donors in methods and materials part.
Question 7: Typing error: the authors should double check the number of enrolled patients. In abstract part, the total number is 695, but in method part, the number is 690.

Reviewer 2 ·

Basic reporting

The authors describe and compare detection of CTCs with classical EpCAM-based approach and CSV-based method in order to capture more mesenchymal CTCs.
The use blood of different cancer patients and of healthy persons.
For a small subset they test both antibodies in the the same blood sample. The observe different positivity rates for both approaches in different tumor entities with CSV slightly outperforming EpCAM.

- language good: only small mistakes
- literature: sufficient
- structure: good, tables are large and complicated and may be placed in a supplement, Figures are small and hard to read.
- discussion is a bit redundant to introduction
- please change the abstract. too many details

Experimental design

- is within scope
- question is well defined
- instead of discussing the potential reason of false-positive "CTCs" the authors could have verified tumor origin by single cell aCGH or low pass sequencing. It would anyway be good to have executed this on a few EpCAM and CSV-positive CTCs. In addition, comparison of such cells from the same patient may point towards different origin of EpCAM and CSV-positve cells.
- antibody clones should be provided

Validity of the findings

- see above (aCGH and low pass)
- there is hardly any statistically significant outcome (it should be marked in the figure)
-

Additional comments

I have made a few comments in the PDF.
- please change the abstract. too many details

Annotated reviews are not available for download in order to protect the identity of reviewers who chose to remain anonymous.

Reviewer 3 ·

Basic reporting

The English in the manuscript is proficient and professional. It is unambiguous for most part except a few lines (line 74, Available should be replaced with discovered/found, line 87-88) Never use etc. (line 63) in a manuscript. Be definite about what you are stating.

Introduction needs more detail. It focuses about EpCAM and CSV, it would be pertinent for authors to describe other CTC markers that are under study and describe why did they chose to focus on these particular markers.

Last section of introduction focuses more on the CytoSorter and that should be a part of the methods section and not introduction.

Figure quality is adequate. In Fig 1, the merged field should always be on the right (last image) by convention and not the first. In Fig 2, the authors need to mention what does the red line signify in Fig 2A and B. Fig D is too small to appreciate the data!

Experimental design

1] The biggest concern I have is that the authors are using a CytoSorter system and for monitoring studies like this a flow based approach is highly adopted for you run a high number of cells to give a high power for statistical significance. Stating number of CTCs, in each sample lacks context. How many cells were actually analysed, what is the percentage in those cells etc.

2] Staining for CSV and the EpCAM should have been performed on the same aliquot together so that would have answered:
a] Are these markers exclusively expressed on cells or have a concurrent expression? Figure 2D would have made more sense if they would have done a flow and then measured it.
Why flow wasn’t used for this study?

3] Authors should have first established the EpCAM and CSV expression in different cancers through TCGA. ( A figure pertaining to it would be nice). That builds up the base for the manuscript, which it currently lacks apart from the explanation in the discussion section. Is there a correlation with EpCAM and CSV expression ?

Validity of the findings

Line 159-162 and 173 – 174 does not inspire confidence in me. “Healthy patients have less than 1 and Cancer patients usually have more than 2 CTCs detected”. I infer that many patients have also 0 or 1 CTC detected the same as healthy! There is not much of a difference between the healthy and cancer. Since, the average EpCAM detection rate is ~62% and ~70% for CSV , the authors wouldn’t be able to confidently tell the healthy and patient apart using this methodology if given a sample blind folded. Again, this would have been overcome if a flow based approached would have been used.

Line 167-168 has data not shown! To be open to the scientific community, I urge the authors to share the data.

Additional comments

It is an extensive study with many different tumor types and numbers.

---

## Round 0.2 · Minor Revisions

Some of the corrections mentioned in the rebuttal letter were not implemented in the manuscript. Please check the annotated version attached and introduce the requested changes before re-submitting.

---

## Round 0.3 · accepted · Accept

You have addressed all concerns in a satisfactory manner.